# Inhibition of PHB1/PHB2 suppresses atherosclerotic plaque formation by interrupting PI3K/AKT/mTOR signaling

Mei Li[1,2]©, Xiaoyan Hu[1]©, Xinxin Hu[1], Fuhua Gao[1], Ying Cui[3,4], Xiaoqing Wei[3,4], Yuanhua Qin[5], Xiaohua An[6], Ying Zhao[3,4]*, Ying Gao ⓘ [1,3,4]*

1 Department of Biochemistry and Molecular Biology, College of Basic Medical Sciences, Dalian Medical University, Dalian, China, 2 Institute of Cancer Stem Cell, Dalian Medical University, Dalian, China, 3 Liaoning Provincial Core Lab of Medical Molecular Biology, Dalian Medical University, Dalian, China, 4 Molecular Medicine Laboratory, Dalian Medical University, Dalian, China, 5 Department of Parasite, College of Basic Medical Sciences, Dalian Medical University, Dalian, China, 6 Clinical Laboratory, Dalian Central Hospital, Dalian, China

© These authors contributed equally to this work.
* gaoying200018@126.com (YG); zhaoying20001105@126.com (YZ)

## Abstract

Prohibitin 1 (PHB1) and prohibitin 2 (PHB2) are highly conserved proteins belonging to the stomatin-prohibitin flotillin-HflC/K (SPFH) protein superfamily. They are ubiquitously expressed and implicated in the regulation of cell proliferation, migration, and survival. However, the expression and biological functions of PHB1/PHB2 in atherosclerosis (AS) remain unclear. In the present study, an enzyme-linked immunosorbent assay was used to detect PHB1/PHB2 expression in the serum of patients with hyperlipidemia. The potential effect and mechanism of PHB1/PHB2 in apolipoprotein E-deficient ($ApoE^{-/-}$) mice were also investigated. shRNA-PHB1 and shRNA-PHB2 lentiviruses were engineered and tail vein-injected into $ApoE^{-/-}$ mice fed a high-fat diet. IL-8, a proatherogenic cytokine, was used as an inducer in vitro. The effects of a PHB1/PHB2 knockdown on vascular smooth muscle cell (VSMC) proliferation, migration, and autophagy and endothelial cell (EC) adhesion were evaluated using methyl thiazolyl tetrazolium (MTT), Transwell migration, Boyden chamber, and monocyte adhesion assays, as well as transmission electron microscopy. Compared with the healthy subjects, PHB1/PHB2 expression was elevated in the serum of patients with hyperlipidemia. Animal experiments showed that downregulation of PHBs reduced the area of atherosclerotic lesions, and the expression of cyclinD1, MMP9, and LC3. In addition, in vitro experiments showed that downregulating PHB1/PHB2 expression under inflammatory stimulation reduced the adhesion, proliferation, migration, and autophagy of ECs and VSMCs by inhibiting the PI3K/Akt/mTOR pathway activation. Collectively, our findings showed that PHBs are activly associated with AS progression.

**Data availability statement:** All relevant data are within the paper.

**Funding:** National Natural Science Foundation of China (No. 30470394, 31070719, 31370800, and 81402916).

**Competing interests:** The authors have declared that no competing interests exist.

## Introduction

Atherosclerosis (AS) is a multifactorial, chronic inflammatory disease and one of the leading causes of death worldwide. Several pathological conditions, such as hypertension, dietary habits, lipid metabolism disorders, and diabetes, ultimately result in AS [1].

A large number of studies have shown that AS occurs within the arterial vessel wall, with inflammatory reactions in the blood vessel being critical to its onset [1]. The aorta is a well-organized, multilayered structure comprising several cell types, mainly, endothelial cells (ECs) and vascular smooth muscle cells (VSMCs). Both the morphogenesis and homeostasis of the blood vessel wall involve highly regulated processes of cell proliferation, migration, and differentiation [2]. During the initiation of AS, endothelial injury or accumulation of low-density lipoproteins (LDLs) triggers the upregulation of cell adhesion molecules, such as vascular cell adhesion molecule-1 (VCAM-1) and intercellular adhesion molecule-1 (ICAM-1) [3]. Concomitantly, the abnormal proliferation and migration of VSMCs also contribute to pathological changes in AS [4]. As an inflammatory disease, AS is the most common underlying pathological cause of coronary artery disease, peripheral artery disease, and cerebrovascular disease. Vascular occlusive plaques accumulate chronically in the subendothelial intimal layer of large arteries, resulting in vascular lumen stenosis, ultimately limiting blood flow and causing severe hypoxia. To date, clinical guidelines have mainly focused on the treatment of AS complications, with clinical methods for inhibiting its progress being limited to the drug-mediated reduction in LDL and cholesterol levels. Therefore, finding an effective therapeutic target for inhibiting AS development is urgent.

Cytokines, which contribute to vascular disease progression by stimulating angiogenesis, migration, and proliferation of cells, are reportedly involved in AS. Interleukin-8 (IL-8), a member of the CXC chemokine family, acts as a chemoattractant for neutrophils and lymphocytes. Interestingly, the proinflammatory cytokine IL-8 has emerged as a key factor in cholesterol homeostasis and AS-associated chemotaxis [5].

Multiple mechanisms may underly the atherosclerotic actions of IL-8, including direct effects on ECs, macrophages, and VSMCs [6]. Upon receiving inflammatory stimuli, IL-8 is upregulated in several vessel wall cells. Thus, IL-8 inducing AS may help reveal novel therapeutic agents to treat AS.

Prohibitins (PHBs) are highly conserved ubiquitously expressed proteins. They include two homologous proteins, namely prohibitin 1 (PHB1) and prohibitin 2 (PHB2), belonging to the stomatin-prohibitin flotillin and HflC/K (SPFH) protein superfamily [7]. PHB1 and PHB2 are encoded by the *PHB1* and *PHB2* genes, located on chromosomes 17 and 12, respectively. PHBs were originally discovered as antiproliferative genes, hence their name. Since then, they have been demonstrated to prevent the growth of specific cells [8]. Despite their possible suppressor role, other studies have indicated that PHBs participate in cell growth and metastasis [9]. PHB1/PHB2, scaffold proteins mainly located in the cytoplasm, mitochondria, nucleus, and plasma membrane, elicit varying functions according to their cellular localization and cell type. In recent years, PHB1 and PHB2 were reported to play different functions in various diseases. However, the role of PHB1/PHB2 in vessel wall cells remains controversial [10]. PHBs consist of an N-terminal transmembrane domain, a conserved PHB domain, and a C-terminal coiled-coil domain, through which the two PHBs mutually interact. PHBs are involved in the PI3K-Akt, MAPK-Erk, and STATS signaling pathways in cell/tissues, which are associated with metabolic regulation and immune functions. The PHB1/PHB2 complex plays an important role in regulating survival, cell cycle progression, and cell proliferation. However, their role and function in AS are poorly characterized.

In this study, we detected PHB1/PHB2 expression in the serum of patients with hyperlipidemia using ELISA. Concurrently, we detected PHB1/PHB2 expression in an AS animal

model. Elevated expression and secretion of PHB1/PHB2 were observed after the exposure of HUVECs and VSMCs to IL-8 inducers. The role of PHB1/PHB2 in the regulation of cell adhesion, proliferation, migration, and autophagy through the Akt pathway was also investigated. Therefore, our present study aimed to elucidate the possible functional role of PHB1/PHB2 in AS signaling and its effects on AS-associated responses.

## Method

### Collection of human serum samples

The human serum samples were collected from Dalian Central Hospital (Dalian, China, 2020.9-2021.1). The serum samples were obtained from 59 hyperlipidemia participants (age range, 33-92 years). Additionally, control serum samples were obtained from 16 health (age range 20-75 years). The hyperlipidemia criteria included: mainly plasma low-density lipoprotein cholesterol (LDL-C) levels $> 3.12$ mmol/L, triglyceride levels $> 1.70$ mmol/L. None of the study subjects were smokers or drank alcohol. This trial was reviewed and approved by the ethics committee of Dalian Central Hospital (Dalian,China)(No.YN2020-031-01). The need for consent was waived by the ethics committee.

### Enzyme-linked immunosorbent assay

Frozen human serum samples and cell's supernatant were thawed for quantification of PHB1 and PHB2 levels with an enzyme-linked immunosorbent assay kit from Shanghai Lengton Bioscience Co.,LTD.

### Lentiviral vector production

To silence PHB1 and PHB2 expression, lentiviral shRNA vectors were constructed using shRNA sequences against PHB1 and PHB2 including GCGGCAACATTTGGGCTTATC(PHB1), CCACATCACAGAACCGAATCT(PHB2). A scrambled control of shRNA lentiviral vector was also constructed using target sequences ACAGAAGCGATTGTTGATC (FulenGen, Shenzhen, China). Lentiviral vectors were produced in HEK293 cells as previously described[28]. Virus titers was $4 \times 10^7$ TU (transduction units)/mL as determined by examining green fluorescent protein (GFP). To screen the target for the most effective gene knockdown, transduced PHB1 and PHB2 were collected for western blot following transduction. For PHB1 overexpression analysis, PHB1 was PCR amplified from cDNA purified from the HUVEC cell line. PHB1 fragment was cloned into pGEX-4T-3 (AmershamBiosciences) plasmid using TOPO technology.

### Animal experiments

This study was conducted in accordance with the Guide for the Care and Use of laboratory Animals published by the US National Institutes of Health (8th edition,2011). The animal protocol was approved by the local research ethics review board of the Animal Ethics committee of Dalian Medical University, and the use of animals and the experimental protocol were approved by the institutional animal care and use committee (IACUC) of Dalian Medical University Laboratory Animal Center. 60 *ApoE-/-* mice (six- week- old males) were purchased from Liaoning Changsheng biotechnology co., Ltd. These mice were randomly divided into four groups (n = 15/group) as follows: (1) Normal diet (ND), (2) Hight-fat diet + empty lentivirus group (HFD), (3) Hight-fat diet + PHB1 lentivirus group (shPHB1), (4) Hight-fat diet + PHB2 lentivirus group (shPHB2). The high-fat diet (MD12015) was purchased from Medicience Ltd. The composition of the high-fat diet was 20% protein, 50% carbohydrate, 21% anhydrous milk fat, and 0.15% cholesterol. Two weeks later, a 200 μl suspension ($4 * 10^7$

TU PHB1 and PHB2) was injected into each mouse through the tail vein twice. Four weeks after transfection, mice were anesthetized with sodium pentobarbital (50 mg/kg) and sacrificed by cervical dislocation. The aortas excision technology was established as previously described[25,26], mouse aortas were excised, and samples of the aortic root were fixed with 4% paraformaldehyde and paraffin-embedded for sectioning. Sections were subjected to hematoxylin/eosin (H&E), Oil red O, Masson Trichrome Staining and immunohistochemical staining.

## Immunohistochemical staining for tissues

The slides were deparaffinized and hydrated with dimethylbenzene and ethanol, and antigens were retrieved and heated in the thermostatic water bath, followed by incubation with goat serum, and then incubated with primary antibodies, anti-PHB1 (Proteintech, China), anti-PHB2(Proteintech, China), and α-smooth muscle actin antibodies overnight at 4 °C. Then slides were the incubated with rabbit anti-mouse (Proteintech, China) for 1h at 37°C. Immunohistochemical staining reactions were visualized using hematoxylin. The images were acquired using a microscope (BX-51, TR32000, Olympus, Japan) and performed the quantification analysis by Image J.

## Oil Red O staining

For the en face analysis of the aorta, the entire aorta and aortic arch were all fixed with 4% paraformaldehyde, and stained with Oil Red O, then opened longitudinally. Results were expressed as percentages of the lipid-accumulating lesion area to the total aortic area analyzed. Image analysis system was used for quantitative measurements (Image J).

## Masson trichrome staining

Samples were in Bouin, then baked for 2h at 60°C, and washed with running tap water, differentiated several seconds with acid alcohol differentiation solution, and rinsed with running tap water for 10 min. Then, samples were immersed in fuchsine acid/Ponceau xylidine for 10 min. Followed by incubating in phosphomolybdic acid for 10 min, in aniline blue solution for 5 min, these samples were briefly washed with $H_2O$, incubated in acetic acid (0.1%) for 2 min, briefly in 95% ethanol. After being dehydrated with 100% ethanol for three times, these samples were made transparent by xylene.

## Cell culture

The rat A7r5 cell line was obtained from American Type Culture Collection (Marassas, VA,USA). Cells were maintained in Dulbecco's modified eagle medium (DMEM; Gibco, USA) at 37°C in a humidified 5% $CO_2$ incubator. The human Aortic Smooth Muscle Cell (AosMC). Cell line was obtained from (American Type Culture Collection, USA). Cells were cultured in DMEM-F-12 medium (Hyclone, USA) in a humidified incubator with 5% $CO_2$ and 95% air at 37°C. The human umbilical vein endothelial cells (HUVECs) line was purchased from American Type Culture Collection. Cells were cultured in high glucose DMEM medium (Hyclone, USA) containing Hepes (Ameresco, USA). The human monocytic cell line (THP-1) was obtained from Cell Bank of Type Culture Collection Committee of Chinese Academy of Sciences, and cells were cultured in RPMI 1640 medium (Gibco, USA). All media were supplemented with 10% fetal bovine serum (FBS; AusGenex, Australia) and 1% penicillin/streptomycin in a humidified incubator with 5% $CO_2$ and 95% air at 37°C. The Akt activator Sc79 (HY-18749) was purchased from MedChemExpress in Shanghai, China.

## Cell transfection

SiPHB1 and siPHB2 were purchased from Genpharm. Transfection was performed using Lipofectamine 2000 (Invitrogen) according to the manufacturer's instructions. NC sense: 5'-UUCUCCGAACGAACGUGUCACGUTT-3', NC antisense: 5'-ACGUGACACGUU CGGAGAATT-3', si-PHB1-641 sense: 5'-GGAUGACGUGUCCUUGACATT-3', si-PHB1-641 antisense: 5'-UGUCAAGGACACGUCAUCCTT-3', si-PHB1-722 sense: 5'-AGCAGAGA GGGCCAGAUUUTT-3', si-PHB1-722 antisense: 5'-AAAUCUGGCCCUCUCUGCUTT=3', si-PHB1-812 sense: 5'-GCUGAUUGCCAACUCACUGTT-3', si-PHB1-812 antisense:5'-CAGUGAGUUGGCAAUCAGCTT-3', si-PHB2-663 sense: 5'-GGACGAUGUAGCUAU CACATT-3', si-PHB2-663 antisense:5'-UGUGAUAGCUACAUCGUCCTT-3'.

## Western blotting

After treatment, cells were lysed in RIPA lysis buffer (50mM Tris-HCl, pH7.4, 150mM NaCl,1% TritonX-100, 1% sodium deoxycholate, 0.1% SDS, 1mM PMSF) on ice and centrifuged at 12000g for 30min at 4°C. The protein concentrations were determined using a BCA protein assay kit (Beyotime Biotechnology, China). Total proteins were separated by electrophoresis on sodium dodecyl sulfate polyacrylamide gels (SDS-PAGE), and then transferred onto polyvinylidene fluoride (PVDF) membranes (Millipore, Billeriea, MA, USA). The membrane was incubated for 1h in 5% milk (w/v)/1%TBS-tween20 solution at room temperature. After blocking membranes were incubated with appropriate dilution of primary antibodies overnight at 4°C. After 3 washes (10min each) with TBST, the membranes were incubated for 1h with horseradish peroxidase-conjugated (HRP-conjugated) secondary antibodies (at appropriate dilution) at room temperature. All the antibodies concentration used in Western Blot, Immunoflurescence are as followed in Table 1:

**Table 1. The antibodies concentration.**

| Antibody | Brand | Dilution ratio | | |
|---|---|---|---|---|
| | | WB | IHC | IF |
| PHB1 | Proteintech | 1:2000 | 1:100 | 1:100 |
| PHB2 | Proteintech | 1:6000 | 1:200 | 1:200 |
| α-SMA | Proteintech | | 1:200 | |
| α-SMA | Sigma | | | 1:200 |
| CyclinD1 | Proteintech | 1:50000 | | |
| MMP9 | Proteintech | 1:1000 | 1:100 | |
| LC3 | Proteintech | 1:2000 | 1:100 | |
| α-tubulin | Proteintech | 1:6000 | | |
| VCAM-1 | Proteintech | 1:2000 | | |
| ICAM-1 | Proteintech | 1:2000 | | |
| Beclin1 | Proteintech | | 1:4000 | |
| p-PI3K | | | | |
| p-AKT (S473) | CST | "1:1000 | | |
| p-AKT (T308) | CST | "1:1000 | | |
| AKT | CST | "1:1000 | | |
| p-mTOR (Ser2448) | CST | "1:1000 | | |
| mTOR | Proteintech | "1:6000 | | |

## Immunoprecipitation assay

Cells were prepared with protein lysis buffer (Beyotime, China). Protein concentrations were determined using the BCA kit (Beyotime, China). Protein lysate (500 µg) was incubated with anti-PHB2 antibody (Proteintech Group, Inc, China) at 4°C for 2h. Then, the mixture was incubated with protein G Plus-Agarose (Sangon Biotech, China) at 4°C overnight. The yielded precipitates were dissolved by 25µL 2x loading buffer, after lysis and centrifuging, and the supernatant was collected for western blotting.

## Immunofluorescence assay

A7r5 cells were grown on 6-well plates with circle microscope cover glass. 70% cells were fixed with 4% paraformaldehyde (20 minutes), permeabilized with 0.5% Triton X-100 (15 minutes), and blocked with 5% BSA (30 minutes). Cells were then incubated with the diluted primary antibodies at 4°C overnight. Next, cells were washed with PBS and incubated with diluted secondary antibodies. Finally, cells were washed with 1xPBS, immersed in DAPI, and stored at 4°C until further use. Images were acquired using an objective lens equipped LeicaTCS SP5II Confocal Laser Scanning Microscope.

## Adhesion of leukocytes to HUVECs (monocyte adhesion assay)

$1x10^5$ cells/mL HUVECs which were transfected with PHB1 plasmid and siRNA, were grown in Hepes medium containing 10% PBS on 96-well plates. They were pretreated with IL-8 (50ng/mL) for 24h. THP-1 monocytic cells were labeled with 2',7'-bis-(2-carboxyethyl)-5-(and-6)-carboxyfluorescein, acetoxymethyl ester (BCECF-AM, Beyotime Biotechnology, China) via incubation for 1h at 37°C. Subsequently, HUVECs were incubated with labeled THP-1 cells for 1h at 37°C。 After being incubated, non-adherent cells were removed by gentle washing twice with PBS and image of random fields was captured using the IX71 microscope (Olympus, Japan).

## Cell proliferation assay

Cell proliferation was measured by two methods: methyl thiazolyl tetrazolium (MTT), and 5-ethynl-2'-deoxyuridired (Edu).

MTT was used to measure the cell viability and proliferation. Pre-transfected A7r5, AosMC were seeded in a 96 well microplate and were cultured with 100µL DMEM or DMEM-F-12 supplemented with 10% PBS. They were cultured for 24h to allow adhere. A7r5, AosMC were then treated with 50ng/mL or serum-starved medium for 24h; cells were incubated with methyl thiazolyl tetrazodium (MTT) for 4h at 37°C in a humidified chamber containing 5% $CO_2$, and then dissolved into 150µLDMSO. Subsequently, optical density (OD) was measured at 570nm using a microplate reader (Thermo Scientific, USA).

Edu cell proliferation Assay (Beyotime Biotechnology, China) was used to measured DNA synthesis. Pre-transfected A7r5 cells were seeded at a $1x10^4$ cells pre well in a 96 well microplate and cultured for 24h. The culture medium was then removed, and cells were incubated in serum medium with $10\,\mu M$ Edu, followed by 50ng/mL IL-8 treatment for 6h. The cells were washed three times with phosphate-buffer saline (PBS) and fixed in 100µl 1% TritonX-100 for 10min at room temperature. After removing TritonX-100, the cells were rinsed twice with PBS, then 50µL Click Reaction (430µL Click Reaction buffer,20µL $CuSO_4$,1µL Azide 488, 50µLClick Addition Solution) was added for 30min protected from dark. After rinsing three times with PBS, nuclei were visualized by Hoechst 33345 staining. Edu-positive cells were imaged under a fluorescence microscope.

## Migration assay

The migration assay was performed using Boyden Chamber and transwell chamber. AosMC cell migration was performed in Boyden Chambers (Neuro Probe, Inc., Gaithersburg, MD, USA) using collagen coated 8μm pore size polycarbonate filters as previously described[25]. Briefly,$1x10^4$ cells were seeded in upper wells of the chamber, which were serum-free DMEM-F-12 containing 0.5% bovine serum albumin. The lower chamber was filled with DMED-F-12-BSA containing 50ng/mL IL-8. Cells were allowed to migrate for 5h at 37°C, 5% $CO_2$. At the end of the assay, cells on the lower filter surface were fixed with methanol and stained with Giemsa, and 9 random fields were counted at 200x magnification. The arbitrary value of 100% was given to the based cell migration assessed in the absence of chemoattractant using fluorescence microscope (BX-51, TR32000, Olympus, Japan). AosMC cell migration was determined by transwell migration assay. $3x10^4$ cells/well in 150μL serum-free DMEM were seeded onto the upper inserts of the 24-well plates (8μm pore size. Corning, New York).500μL of the low-serum (2% BSA) medium with or without IL-8 (50ng/mL) was added to the lower chamber. Cells were allowed to migration for 24h at 37°C, 5%$CO_2$. After 24h, migration cells on the lower surface were fixed with methanol and stained with Giemsa for 10 minutes. Cell migration was observed under an optical microscope (BX51, Olympus, Japan)

## Autophagy assay

A transmission electron microscopy analysis (JEOL, Japan) was performed to observe the autophagic vacuoles cells which were harvested with 3000g centrifuging for 4min and fixed in 2.5% glutaraldehyde in PBS overnight. The cells were washed three times with PBS, and the samples were postfixed in 1% osmium tetroxide for 2h and dehydrated in a gradient elute with different concentration ethanol. After removing 100% ethanol, the cells were added propylene oxide for 10min, and kept overnight in Epoxy Embedding Medium. After different temperature curings in an oven, the sections were ultracut. The ultrathin sections were stained with uranyl acetate and lead citrate and then observed using a transmission electron microscope. Autophagosome was identified under the microscope solely based on size and morphology.

## Statistical analysis

The least three repeats were performed for each experiment, and dealt with the data with GraphPad Prism 9.0. Values were presented as mean±standard error of the mean (SEM). Significant differences between the groups weredetermined using the one-way ANOVA.Differences with pvalues less than 0.05 were considered statistically.

# Results

## 1. Secretion of PHB1/PHB2 proteins was increased in the serum of patients with hyperlipidemia

Hyperlipidemia is considered to be a significant risk factor for cardiovascular, especially atherosclerosis. Excessive blood lipids can cause disorders in lipid metabolism and increase blood viscosity and lipid deposits in the intima of blood vessel walls, leading to the gradual accumulation of atherosclerotic plaques. Hyperlipidemia predisposes patients to AS through the entry of LDL-C particles into arterial subendothelial spaces [11]. In this study, we selected patients with high serum LDL levels as the hyperlipidemia group. We also evaluated the levels of PHBs in serum samples from healthy controls and patients with hyperlipidemia. We noticed that when comparing assays, we obtained different results. Notably, we found that the secretion level of PHBs in patients with hyperlipidemia was higher than that in healthy subjects (paired

2-tailed $t$-test, $p < 0.05$, for all comparisons) (Fig 1). These results suggested the induction of PHB1/PHB2 in hyperlipidemia, highlighting the requirement to further explore the secretion pathway of PHB1/PHB2. In addition, these results suggested that PHB1 and PHB2 are possibly involved in AS.

## 2. PHB1 and PHB2 expression was increased and affected atherosclerosis plaque formation in $ApoE^{-/-}$ mice

It is known that $ApoE^{-/-}$ mice show impaired clearing of plasma lipoproteins, rapidly developing AS. Hence, we used $ApoE^{-/-}$ mice as animal models of atherogenesis. In order to identify the role of PHB1 and PHB2 in atherogenesis, we established an AS model using $ApoE^{-/-}$ mice fed a high-fat diet. We found that PHB1 and PHB2 expression in atherosclerotic plaque-containing vessels was significantly increased compared with that in normal blood vessels. We also observed that PHB1 and PHB2 were expressed in vascular wall endothelial and smooth muscle cells, mainly in the cytoplasm (S1,S2 Figs). Subsequently, we engineered shRNA-PHB1 and shRNA–PHB2 lentiviral vectors, which were injected into the tail vein of $ApoE^{-/-}$ mice fed a high-fat diet. Western blotting analysis of proteins extracted from aortic tissue demonstrated that PHB1 and PHB2 expression was lower in lentivirally infected shPHBs mice compared with those in the ND (normal diet) group (Fig 2A), indicating the effectiveness of silencing vectors. We performed a morphological observation 8 weeks after feeding $ApoE^{-/-}$ mice a high-fat diet and confirmed the successful establishment of the AS murine model via histopathological analysis using hematoxylin and eosin (H&E) and Oil Red O staining. We used Masson's trichrome-stained tissue sections to quantify the collagen content in each sample. Sections stained with H&E and Oil Red O revealed that atherosclerotic lesions in the lentiviral-infected shPHBs groups were smaller than those in the ND group. Masson's trichrome staining revealed an increased amount of collagen fibers in the thoracic aortic tissues of control mice. By contrast, AS model mice had thinner fibrous caps and fewer collagen fibers in their thoracic aorta. Compared with the model group, mice in the lentiviral-infected shPHBs groups showed homogeneously increased amounts of collagen fibers (Fig 2B and C).

Abnormal proliferation, migration, and autophagy of VSMCs are critical processes involved in AS [12]. We used immunohistochemistry to investigate the effects of AS on

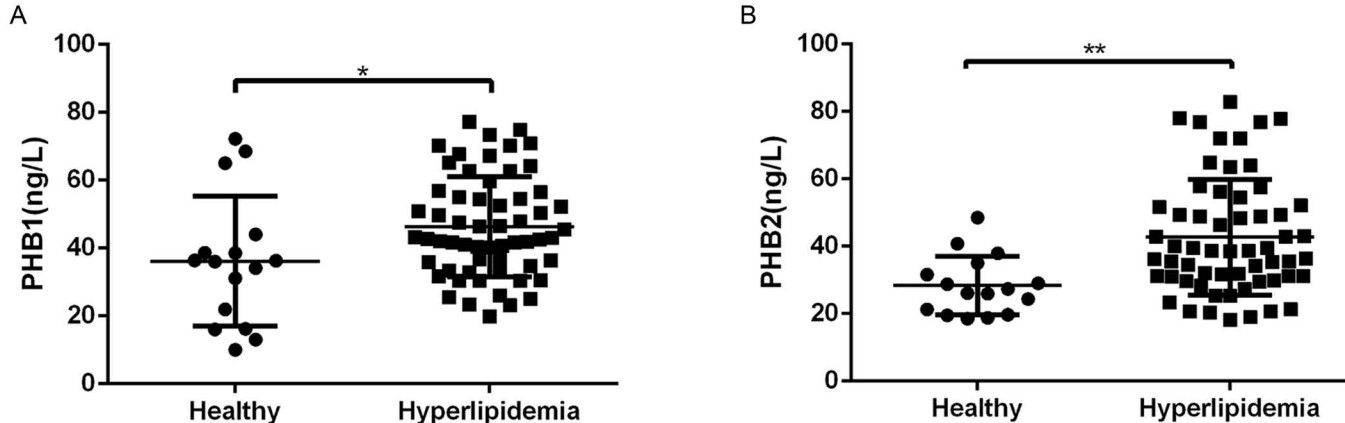

**Fig 1. PHB1/PHB2 protein secretion in the serum of patients with hyperlipidemia.** (A) Level of PHB1 secretion in the serum of patients with hyperlipidemia (n = 59) and healthy controls (n = 16). (B) Level of PHB2 secretion in the serum of patients with hyperlipidemia (n = 59) and healthy controls (n = 16). *$p < 0.05$, **$p < 0.01$.

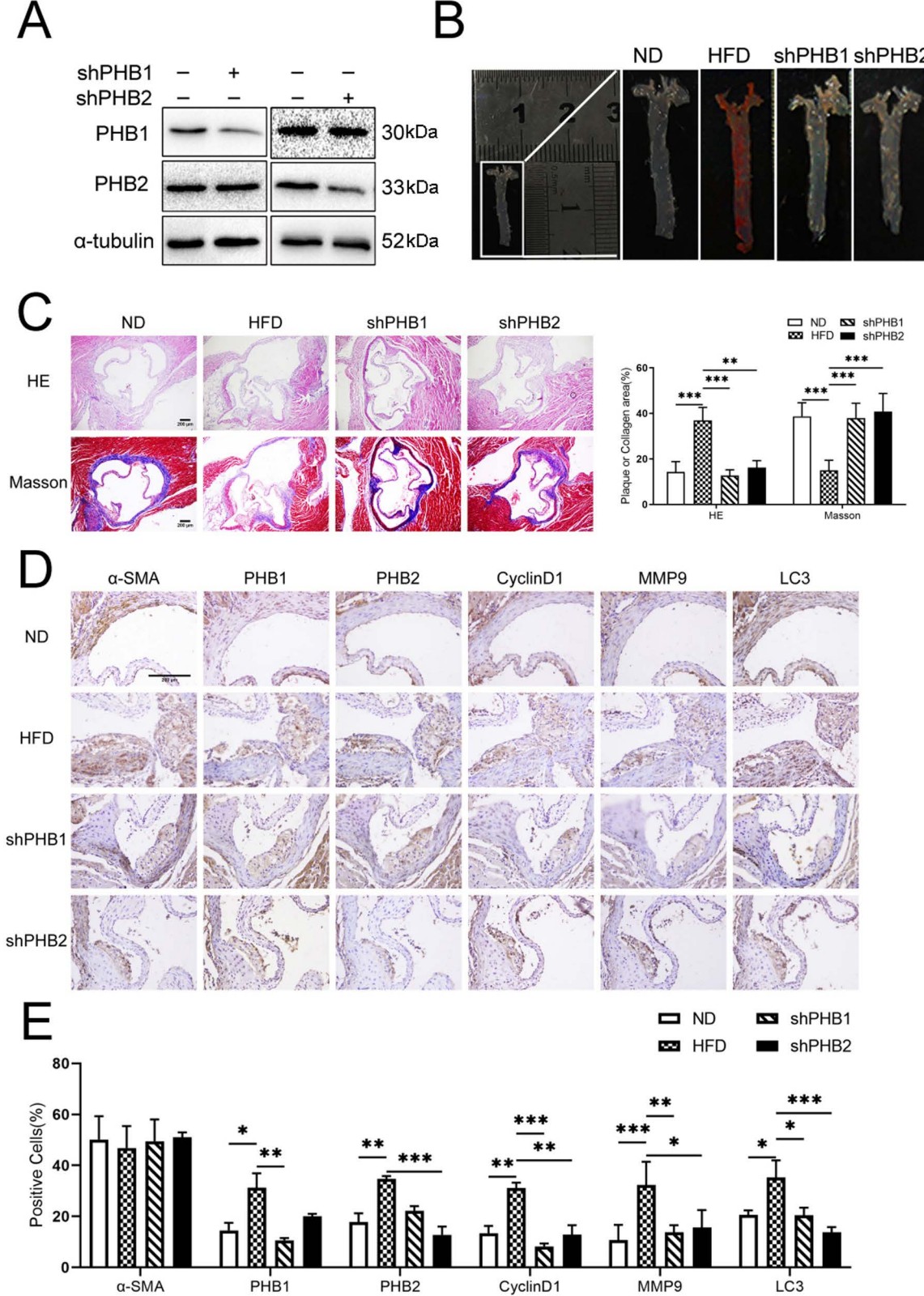

**Fig 2. Effects of lentiviral carriers in *ApoE*⁻/⁻ mice.** (A) PHB1 and PHB2 protein expression in the aortic root in empty lentivirus group and lentiviral-transfected groups of mice as determined by western blotting (n = 5). (B) Oil Red O staining of aortas in four

groups of mice (ND, HFD, shPHB1, and shPHB2). (C) Representative hematoxylin and eosin staining and Masson's staining of aortic plaques (scale bar: 200 μm, n = 6). (D, E) Representative immunohistochemically stained images and quantification of PHB1, PHB2, cyclin D1, MMP9, and LC3 levels in atherosclerotic plaques in four groups of mice (scale bar: 200 μm, n = 6). Data are expressed as the mean ± SD. *$p < 0.05$, **$p < 0.01$, ***$p < 0.001$. Normal diet (ND), Hight-fat diet + empty lentivirus group (HFD).

the expression of PHBs and SMC-α actin in VSMCs. We found that the silencing of PHBs decreased the activity of cyclin D1, MMP-9, and LC3 in VSMCs relative to that in the control group (Fig 2D and E). We also noticed that the PHB1/PHB2 OD values were increased in the AS group compared with those in the ND (control) group (all $p < 0.05$). These results indicated that AS promotes PHB1/PHB2 expression in arterial plaques. Of note, we detected that PHBs were predominantly expressed in the cytoplasm of VSMCs (Fig 2D). Our findings further revealed that PHB1/PHB2 promoted proliferation, migration, and autophagy *in vivo*, indicating that PHB1/PHB2 might increase the formation of vulnerable atherosclerotic plaques *in vivo*.

We did not observe significant differences in the body weight of mice among all groups. The average mice body weight was 24.24 ± 2.200 g, suggesting that the lentiviral-mediated gene knockdown of PHBs did not affect animal growth.

## 3. Expression and secretion of PHB1 and PHB2 in VSMCs and HUVECs

To investigate PHB1/PHB2 expression patterns, we stimulated VSMCs and HUVECs (The human umbilical vein endothelial cells) with a gradient concentration of IL-8 (0, 5, 10, 20, 50, and 100 ng/mL). Accordingly, IL-8 dose-dependently increased PHB1/PHB2 expression at 24 h in VSMCs and HUVECs (S3A Fig). Besides, we observed that when treated with IL-8 (50 ng/mL) for different durations (6, 12, 24, 48, and 72 h), PHB1/PHB2 expression in VSMCs and HUVECs was increased in a time-dependent manner (S3B Fig).

We also stimulated VSMCs and HUVECs with IL-8 to evaluate the secretion level of PHB1/PHB2 during VSMC proliferation. After treating cells with IL-8 (50 ng/mL), we collected the supernatant for ELISA (Fig 3A,B). To confirm the binding between PHB1 and PHB2 at the protein level, we performed a coimmunoprecipitation (CoIP) analysis. Our CoIP assay demonstrated that the expression of PHB1 was downregulated in siPHB2-treated A7r5 cells. In addition, CoIP with anti-PHB2 led to the detection of a PHB1 band, which was markedly stronger than that detected following a control incubation without the anti-PHB2 antibody, confirming that PHB1 and PHB2 physically interact (Fig 3C). The microenvironment in which proteins reside offers ideal conditions for exerting their levels of expression and functions. Therefore, localization drastically impacts protein function. We determined the distribution and colocalization of PHB1 and PHB2 in VSMCs using confocal microscopy and found that PHBs were mainly distributed in the mitochondrial compartments within the cytoplasm (Fig 3D).

## 4. siRNA-PHB1 decreased the IL-8-mediated adhesion of monocytes to ECs

AS starts with the adhesion of inflammatory monocytes on activated ECs in response to inflammatory stimuli [13]. To investigate the role of PHB1 in the adhesion of THP-1 monocytes to HUVECs upon treatment with IL-8, we used 2 different methods followed by PHB1 expression analysis; in the first, we knocked down PHB1 using si-812, and in the second, we overexpressed PHB1 using pex-PHB1. We accordingly found that PHB1 expression in HUVECs was attenuated following transfection with si-812-PHB1, whereas it was increased by pex-4-PHB1 (S4A,B Fig).

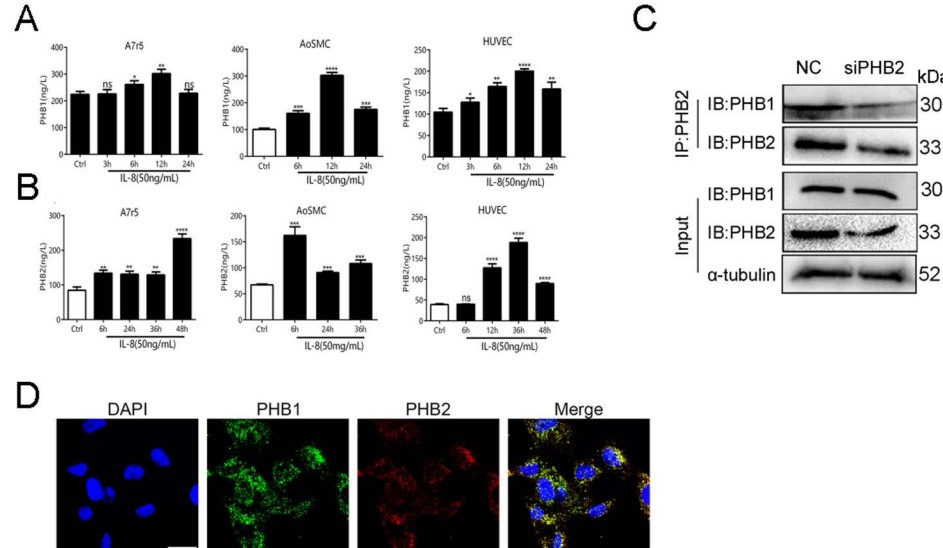

**Fig 3. Secretion of PHB1 and PHB2 in blood vessel wall cells.** (A, B) ELISA assay showing the secretion of PHB1 and PHB2 in HUVEC, A7r5, and AosMC cells upon treatment with IL-8 (50 ng/mL). (C) CoIP assay combined with western blotting for evaluating the interaction between PHB1 and PHB2. (D) Confocal microscopy showed that PHB1 colocalized with PHB2 in the cytosol. Cell nuclei were stained with DAPI (blue); scale bar: 25 μm. All results are expressed as the mean ± SD (n = 5). *$p < 0.05$, **$p < 0.01$, ***$p < 0.001$.

We noticed that the adhesion of monocytes to HUVECs upon treatment with IL-8 was decreased after knocking down PHB1, whereas overexpression of PHB1 did not result in a statistically significant change in monocyte adhesion. (Fig 4A,B). Further investigation through western blotting to detect the expression of the ICAM-1 and VCAM-1 EC adhesion-related molecules, and found that si-PHB1 treatment impaired ICAM-1 and VCAM-1 levels in IL-8-mediated adhesion.

## 5. Downregulation of PHB1 and PHB2 reduced VSMC proliferation

VSMCs are the main cell type of the arterial wall. They have a very low proliferation rate in normal blood vessels but show a high proliferation rate in the early stage of AS and following vascular injury. Studies have shown that the abnormal proliferation of VSMCs plays a key role in the pathogenesis and progression of AS. To further investigate the role of PHBs in VSMCs, we knocked down PHB1 in aortic smooth muscle cells (AosMC), and A7r5 using si-812 and si-722, respectively, and knocked down PHB2 in A7r5 using si-663. As shown in Fig 5A, PHB1 expression was upregulated by pex-4-PHB1 plasmids in AosMCs (Fig 5B). Consecutively, we performed MTT and EdU assays to test the viability and proliferation of IL-8-treated VSMCs. Our MTT assay demonstrated that A7r5 cell viability was increased by IL-8 in a dose-dependent manner (Fig 5C). In addition, we found that A7r5 and AosMC cell proliferation was increased upon IL-8 stimulation, whereas knocking down PHB1 and PHB2 separately or together expectedly inhibited cell proliferation (Fig 5D,E). Our EdU assay illustrated that knocking down PHB1 and PHB2 separately or simultaneously inhibited cell proliferation (Fig 5F). Thus, these results suggested that PHB1 and PHB2 inhibit the IL-8-induced proliferation of VSMCs. Furthermore, knocking down PHB1 and PHB2 repressed the increased expression of cyclin D1 in IL-8-induced A7r5 cells (Fig 5G).

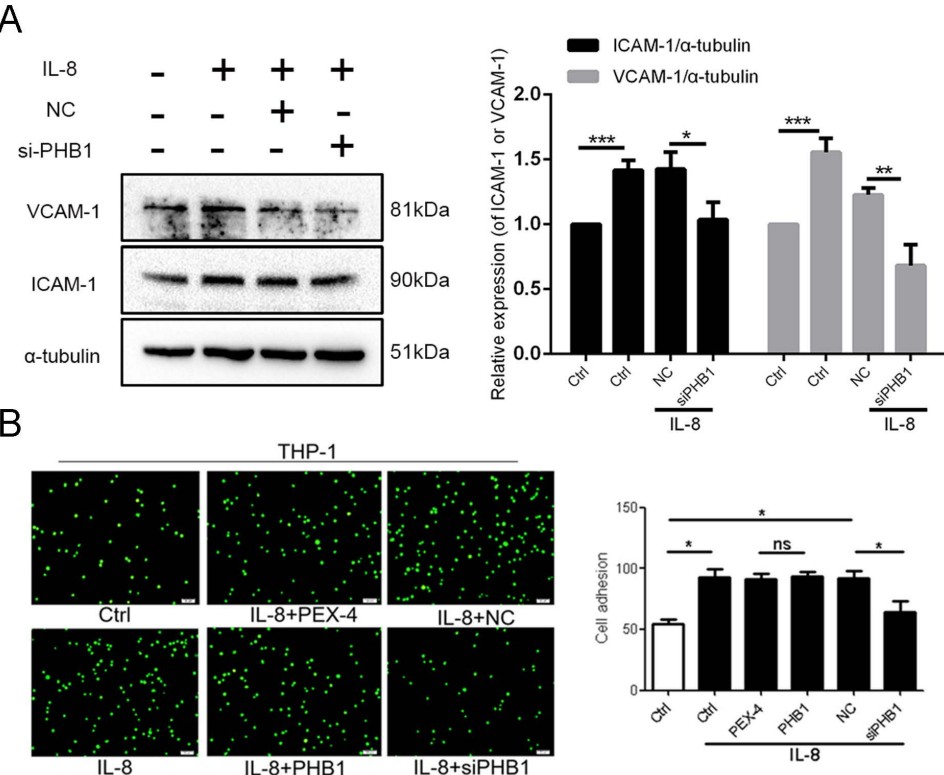

**Fig 4. PHB1 contributes to the IL-8-induced adhesion of monocytes to ECs.** (A) HUVECs subjected to down- or upregulation of PHB1 for 24 h were pretreated with IL-8 (50 ng/mL) and then co-incubated with BCECF-labeled THP-1 cells for 30 min. Representative images and quantitative analysis of the attachment of THP-1 cells. The number of adherent THP-1 cells in nine random fields of view was counted. Graphic shows the statistical data on adhesion; scale bars: 200 μm. (B) The expression of ICAM-1 was determined using western blotting. All data are expressed as the mean ± SD (n = 5). $*p < 0.05$, $**p < 0.01$, $***p < 0.001$.

## 6. PHB1 and PHB2 downregulation reduced VSMC migration

The abnormal proliferation of VSMCs and their migration from the circulation to the subintimal space are key factors in AS pathogenesis. VSMCs are affected by deposited lipids and various inflammatory factors, prompting them to proliferate and migrate toward the intima. Studies on the migration of VSMCs can help clarify the occurrence and development of AS. To determine the effect of PHB1 and PHB2 on the IL-8-induced migration of VSMCs, we performed Boyden and Transwell chamber migration assays. Boyden chamber assay results revealed that knocking down PHB1 markedly suppressed the IL-8-mediated migration of AosMCs (Fig 6A). Similarly, the Transwell assay indicated that PHB2 counteracted the IL-8-induced migration of VSMCs by 61.5%, 65.9%, and 82.1%, respectively ($p < 0.05$) when PHB1, PHB2, or PHB1 and PHB2 were knocked down (Fig 6B). Finally, western blot analysis indicated that PHB1 and PHB2 inhibit the IL-8-induced migration of VSMCs (Fig 6C,D).

## 7. Autophagic vacuole formation was reversed by PHB1 and PHB2 downregulation in IL-8-induced VSMCs

Inflammation is reportedly involved in AS development, and *in vitro* experiments have shown that inflammatory factors activate autophagy, suggesting potential new targets for the treatment of AS [14].

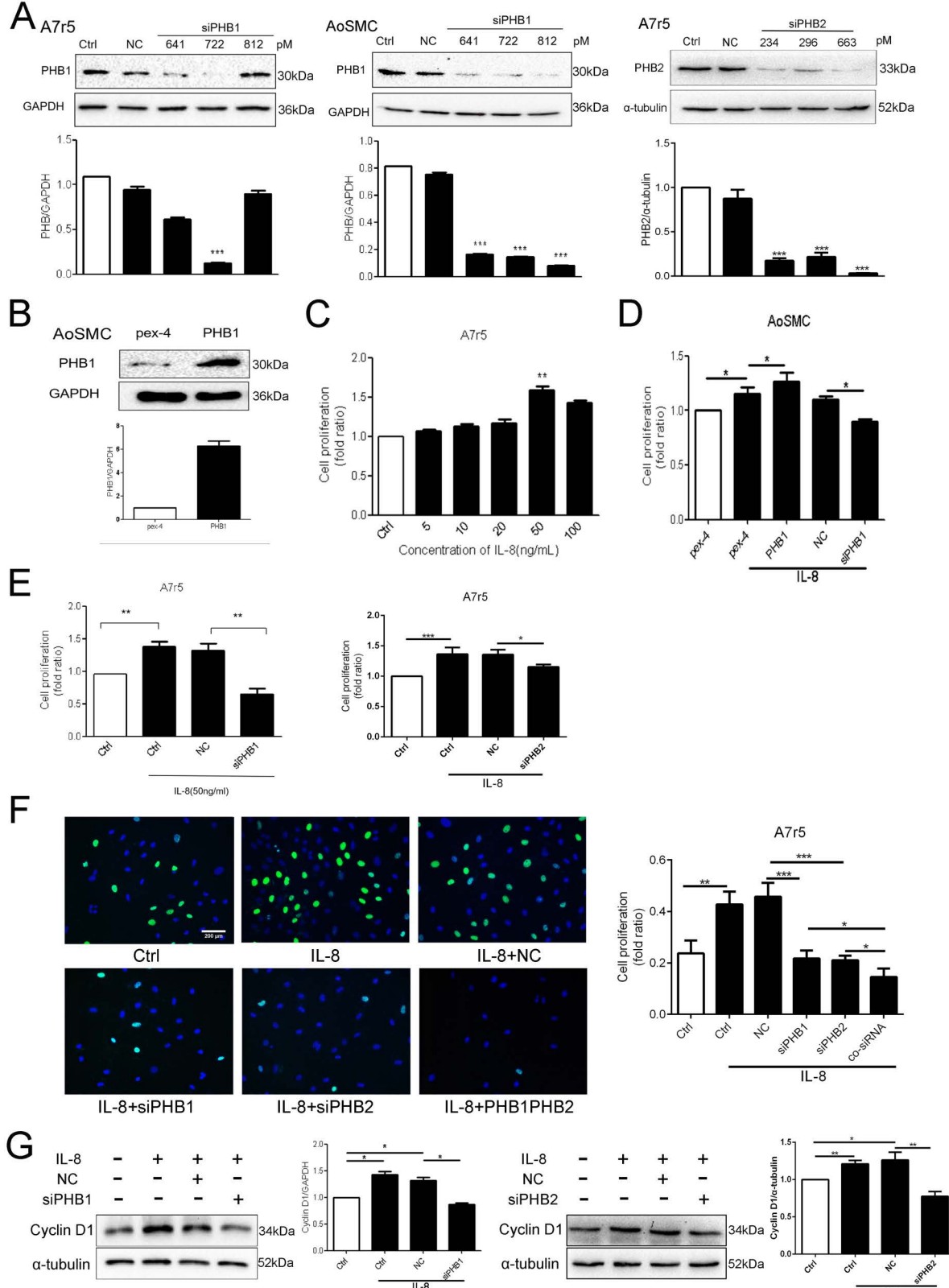

**Fig 5. PHB1 and PHB2 inhibited the viability and proliferation potential of IL-8-induced VSMCs.** (A) Western blotting confirmed the efficiency of siPHB1 and siPHB2 in VSMCs; siPHB1 represents siRNA-PHB1, while siPHB2 represents siRNA-PHB2. NC represents

negative siRNA used as control. (B) Western blotting confirmed the efficiency of the pex-4-PHB1 plasmid in AosMC cells. Pex-4 represents the empty vector, whereas PHB1 represents the pex-4-PHB1 plasmid. (C, D, E) Cell proliferation was measured using the MTT assay. (F) Cell proliferation was measured using the EdU assay; scale bars: 200 μm. (G) Cyclin D1 expression was determined using western blotting. All data are expressed as the mean ± SD (n = 5). *$p < 0.05$, **$p < 0.01$, ***$p < 0.001$.

To determine the effect of PHBs on autophagy, we used transmission electron microscopy to observe autophagic vacuole formation. In contrast to the normal morphology of control cells, we observed typical signs of autophagy in IL-8-treated cells, including the presence of organelle-containing autophagic vacuoles. We observed that PHB1 overexpression activated autophagy in IL-8-treated AosMCs. In contrast, knocking down PHBs in IL-8-induced VSMCs reduced the number of autophagosomes. Likewise, we observed fewer autophagosomes in si-PHB-treated cells under transmission electron microscopy (Fig 7A,B). These results demonstrated that PHBs activate autophagy in VSMCs. Beclin-1 is reportedly involved in the formation of autophagic vacuoles and autophagic cell death [15]. LC3 lies in the membranes of autophagosomes, and conversion of LC3-I to LC3-II is indicative of the increase in autophagic activity [16]. The enhanced conversion of the cleaved LC3-I to LC3-II in our experiments confirmed the effects of PHB1 on autophagy. Western blot analysis showed that the expression of Beclin-1, LC3-I, and LC3-II was increased after IL-8 treatment (Fig 7C,D).

## 8. PI3K/Akt/p-mTOR signaling mediated the IL-8-induced expression of PHB1 and PHB2

Previous studies have demonstrated that the Akt and MAPK signaling pathways are involved in the IL-8-mediated overexpression of PHB1 in VSMCs [17,18], and PHB1/PHB2 play a cellular survival role through MAPK and Akt [19]. To investigate the role of the PI3K/Akt pathway in PHB1- and PHB2-driven proliferation, migration, and autophagy of VSMCs, we evaluated the phosphorylation of Akt and MAPK in response to IL-8 treatment using western blotting. We found that the phosphorylation levels of PI3K, Akt, and mTOR were upregulated in IL-8-treated A7r5 cells compared with those in the ND group, while the total protein levels of PI3K, Akt, and mTOR between the two groups were seemingly discrepant. These findings suggested that knocking down PHB1 and PHB2 in IL-8-treated A7r5 cells results in the gradual reduction in the expression of PI3K, p-Akt (Ser473), p-Akt (T308), and p-mTOR (Fig 8A,B,C, and D). We then used an Akt agonist (SC79) to confirm the involvement of PHB1/PHB2 in the proliferation, migration, and autophagy of VSMCs via the Akt-dependent signal pathway (Fig 8E,F,G, and H). As shown in Fig 8H through 8L, SC79 treatment induced a pronounced activation of the Akt pathway. This activation was evidenced by heightened levels of phosphorylated Akt, concomitant with an upsurge in PHB1 and PHB2 expression, suggesting a regulatory feedback loop that enhances cellular proliferation, migration, and autophagy under inflammatory conditions induced by IL-8. Significantly, the reversal of si-PHB1 and si-PHB2 effects by SC79 treatment highlights the integral roles of these proteins in modulating VSMC behavior through Akt-mediated signaling pathways. The increased autophagic activity, observed as elevated LC3-II levels alongside decreased p62, confirms an enhanced autophagic flux. This is crucial in maintaining cellular homeostasis and responding to inflammatory stimuli, demonstrating that PHB proteins are not merely passive players but active modulators of cellular resilience against atherosclerotic influences. Detailed analysis of these interactions provides valuable insights into potential therapeutic targets within vascular pathophysiology. We also examined the effects of si-PHB1 on the activity of Erk and the p-c-Raf upstream protein. We detected that knocking down PHB1 resulted in a noticeable change in the activity of Erk in IL-8-treated A7r5 cells, whereas no difference was observed in the activity of p-Erk

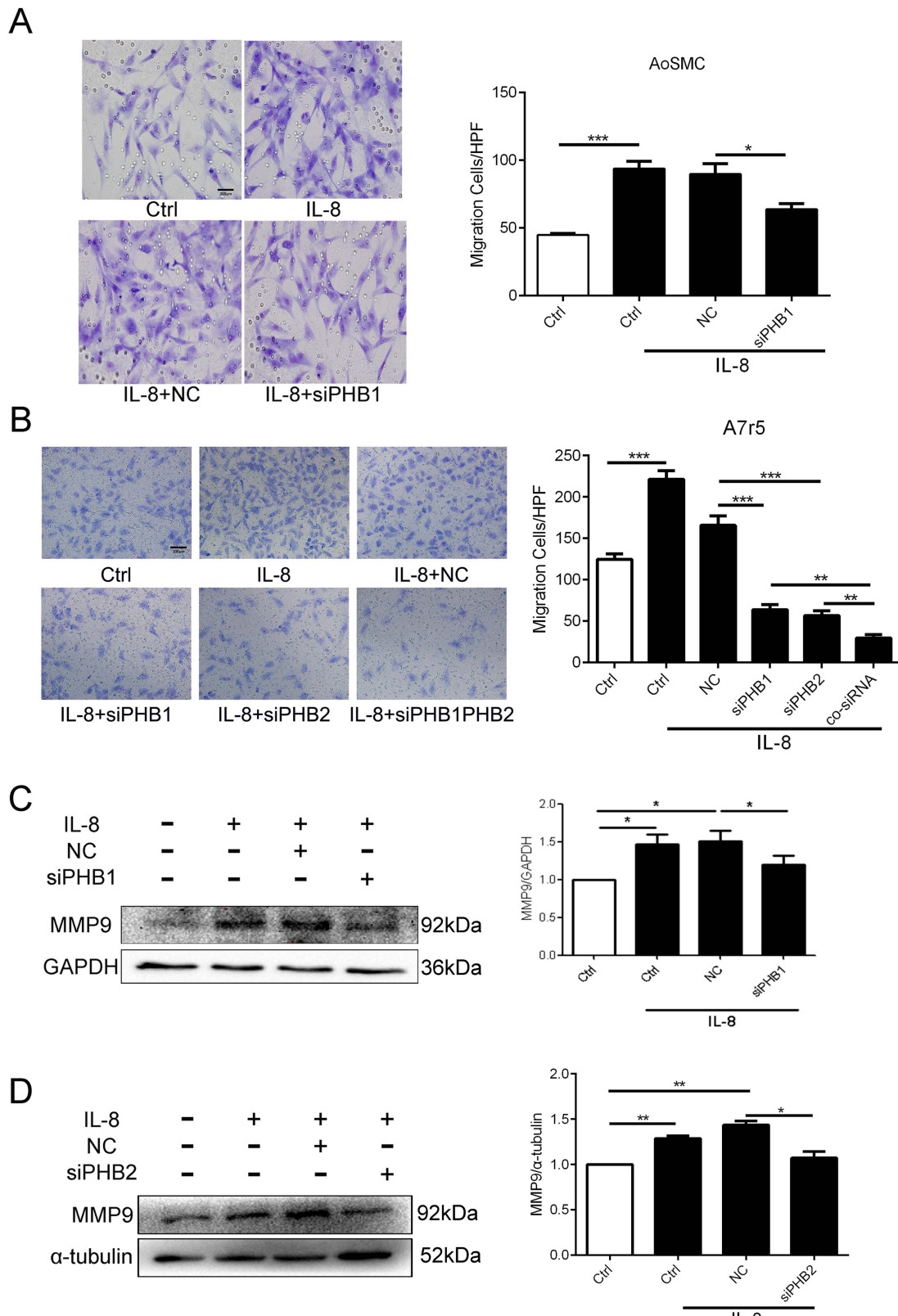

**Fig 6. Downregulation of PHB1 and PHB2 reduced VSMC migration.** (A) Cell migration was measured using the Boyden chamber assay; scale bars: 200 μm. (B) Cell migration was measured using the Transwell assay; scale bars: 200 μm. (C, D) The expression of MMP9 was determined using western blotting. All data are expressed as the mean ± SD (n = 5). *$p < 0.05$, **$p < 0.01$, ***$p < 0.001$.

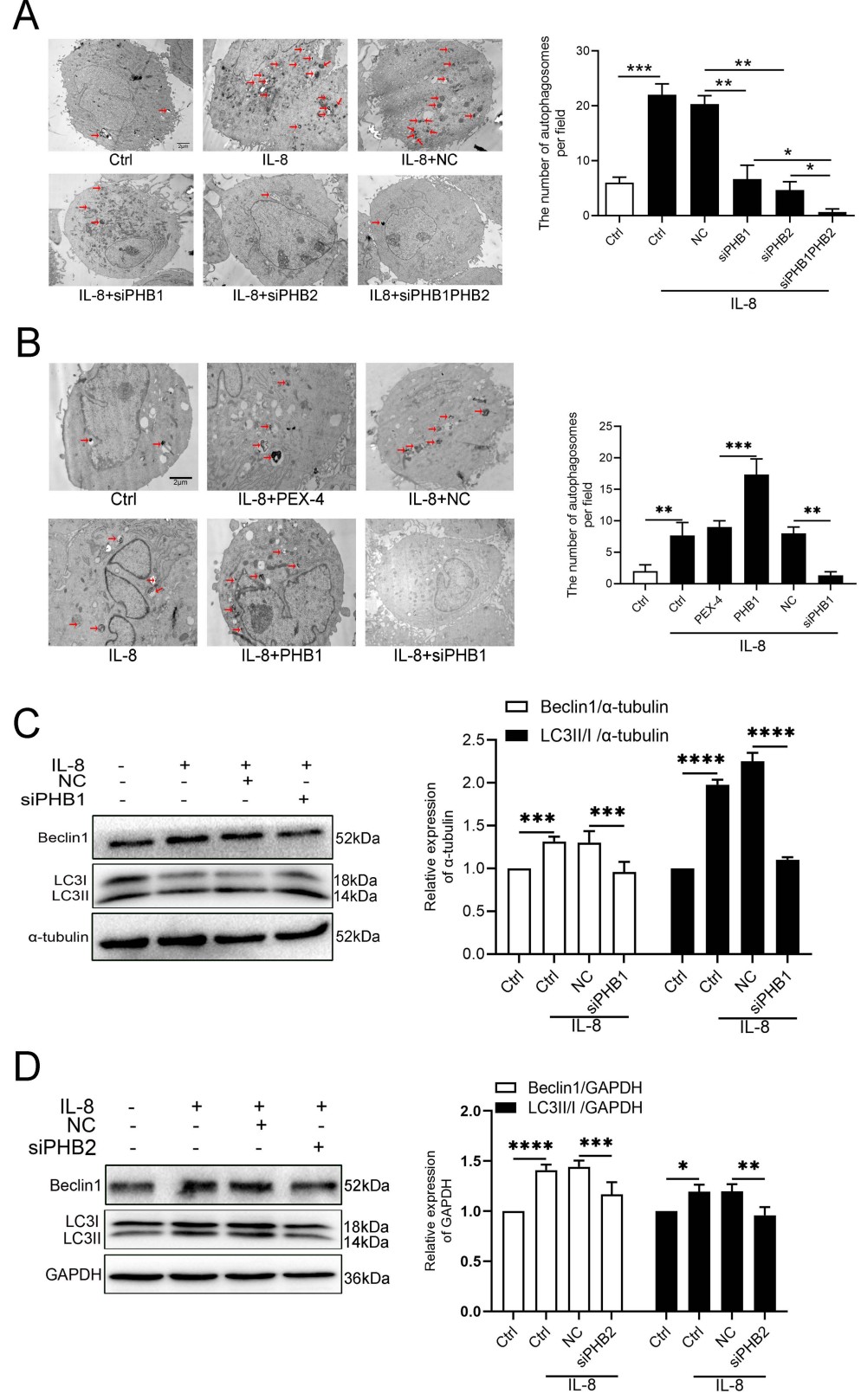

**Fig 7. The formation of autophagic vacuoles was reversed by PHB1 and PHB2 downregulation in IL-8-induced VSMCs.** (A) Autophagic vacuoles in A7r5 cells observed by TEM; scale bars: 2μm; arrows represent autophagosomes.

(B) Autophagic vacuoles in AosMC cells observed by TEM. Representative images are shown; scale bars: 2 μm; arrows represent autophagosomes. (C, D) The expression of Beclin 1, LC3-I, and LC3-II was determined using western blotting. All data are expressed as the mean ± SD (n = 5). *$p < 0.05$, **$p < 0.01$, ***$p < 0.001$, ****$p < 0.0001$.

relative to that in the control. Therefore, we assumed that the Erk signaling pathway was not the main regulatory signaling pathway (S5 Fig).

## Discussion

Prohibitin was initially associated with the inhibition of cell proliferation, hence the naming. As it is reportedly present in different cellular compartments, prohibitin plays a critical biological role in mitochondrial function, cell proliferation, and development [20]. Prohibitin was used as an inducer in rheumatoid arthritis and as a novel autoantigen in previous studies [17], suggesting that PHB1 and PHB2 might play a role as secretory proteins. Both PHB1 and PHB2 contain unconventional non-cleavable targeting sequences on their N-terminal, implying that the PHB complex protein is likely a secretory protein. This was confirmed when we used ELISA to measure the PHB complex in the supernatant and serum.

PHB1 is a pleiotropic protein that functions as a growth regulatory molecule in several tissues and is primarily expressed in adipocytes, immune, cancer, and diabetic cells. PHB2, also called B cell receptor-associated protein 37 (BAP37) and repressor of estrogen receptor activity (REA), is expressed in immune cells in specific diseases such as lipodystrophy, diabetes, immunodeficiencies, and muscular dystrophy/wasting [21]. In addition, PHB2 is often upregulated in cancer and diabetes. The PHB1/PHB2 complex is also present on the surface of platelets and lipid rafts [22]. Dysfunctions of PHB1/PHB2 are known to be associated with podocyte cytotoxicity [23], oxidative stress, and metabolic diseases [24]. PHB1 and PHB2 function not only as a heterodimeric complex but also independently. Several studies have reported that GOLPH3 promotes autophagy in glioma cells via PHB2 but not PHB1 [25]. In our study, we found that PHB1 and PHB2 had the same effect on AS. When PHB1/PHB2 expression was decreased, AS development was inhibited. However, it was reported that PHB1 inhibition reduced oxidative stress and regulated the transduction of the Akt/GSK-3β signaling pathway, providing cellular protection against ISO-induced hypertrophy in H9c2 cardiomyocytes [10]. This differed from the effect of PHB on the cardiovascular system. The possible reasons behind this difference may be that even if the cytoplasmic calcium concentration in cardiomyocytes and smooth muscle cells reaches the corresponding threshold for stimulating contraction, their contraction initiation mechanisms differ; that is, the key calcium-responding proteins in the two cell types are distinct. Myosin C is the key calcium-responding protein in cardiomyocytes, whereas calmodulin and myosin light chain kinase (MLCK) are the key calcium-responding proteins in smooth muscle cells [26]. This difference might be one of the reasons for the distinctive roles of PHB in the two cell types. Our study emphasized that PHB1 and PHB2 might be new targets for treating AS.

Our *in vivo* experiments demonstrated increased serum PHB1/PHB2 levels in patients with hyperlipidemia. Immunohistochemical and immunofluorescence staining confirmed the expression of PHB1/PHB2 on VSMCs in both human and mice aortic vessel cells. PHB1/PHB2 expression in artery plaques in both AS mice and patients was significantly higher than in respective controls. Hence, PHBs might play an important regulatory role in the pathophysiology of AS. Relevant *in vitro* experiments further demonstrated the role of PHB1/PHB2 on the development of AS. Various modifications in retained lipoproteins primarily triggered a low-grade inflammatory response, leading to the activation of ECs, VSMCs, and macrophages. Chronic exposure of ECs to ox-LDL induces an inflammatory response and

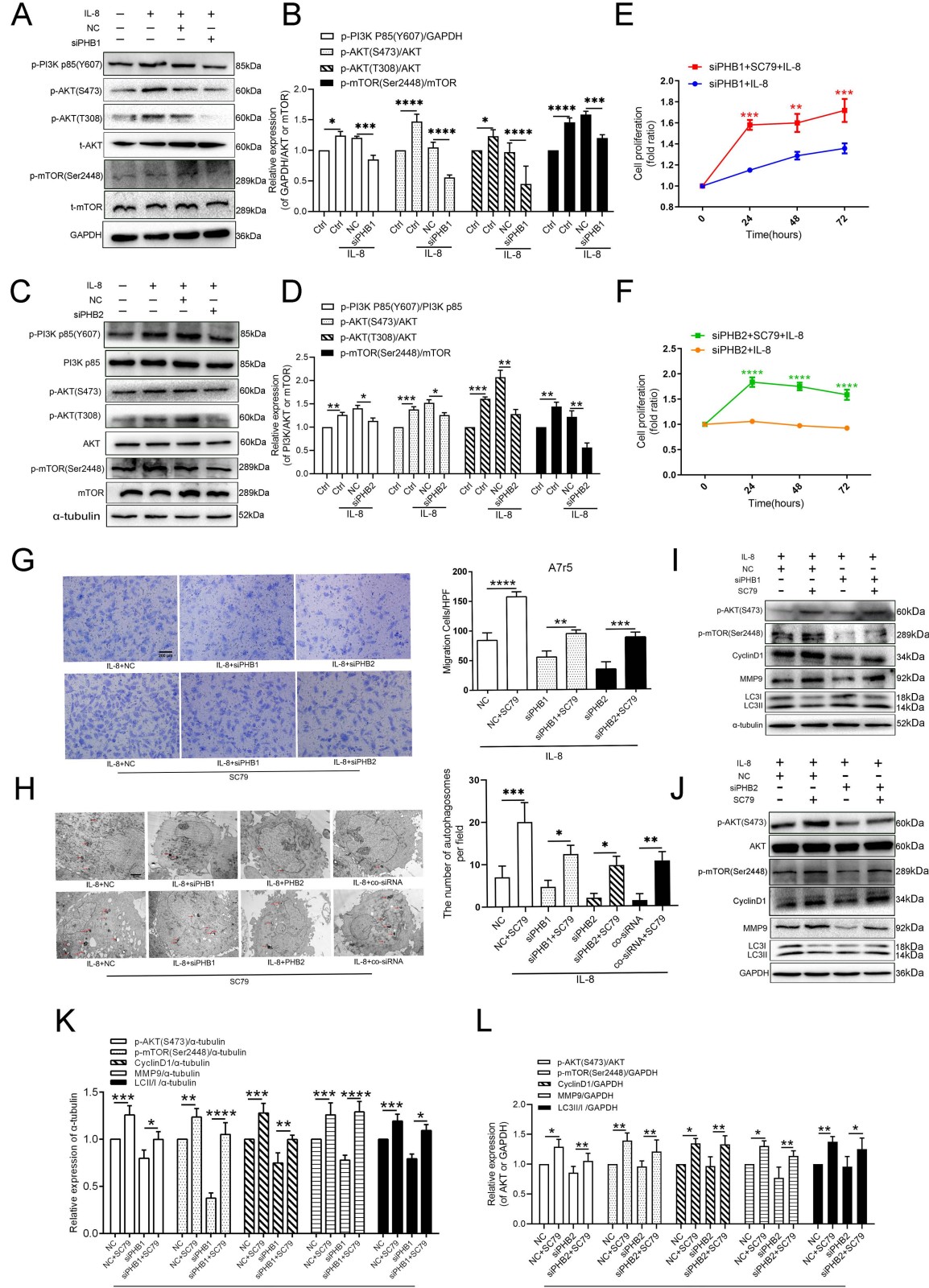

**Fig 8. Effects of the Akt signaling pathway on the IL-8-induced proliferation, migration, and autophagy of VSMCs upon treatment with siRNA-PHB1 and siRNA-PHB2.** (A, C) The activation status of p-Akt (S473), p-Akt (T308), p-PI3K P85(Y607) and p-mTOR(Ser2448) after IL-8 treatment was detected by western blotting. (B, D) Quantification of protein phosphorylation (vs. total

protein). (E, F, G, H) MTT, Transwell, and TEM assays assessing the proliferation, migration, and autophagy of IL-8-induced A7r5 cells treated with SC79 and/or siRNA-PHB1, and siRNA-PHB2. Representative images are shown; scale bars: 200 μm (in G) and 2 μm (in H). (I, J) Western blotting was used to examine the activation status of p-Akt (S473), p-mTOR (Ser2448), cyclin D1, MMP9, LC3-I, and LC3-II in IL-8-induced A7r5 cells treated with SC79 and/or siRNA-PHB1, and siRNA-PHB2. (K, L) Quantification of protein phosphorylation (vs. total protein). All data are expressed as the mean ± SD (n = 5). *$p < 0.05$, **$p < 0.01$, ***$p < 0.001$.

the expression of vascular adhesion molecules [27]. VSMCs, as the main cell type in blood vessel walls, have been extensively studied regarding their proliferation, migration, phenotype switching, response to oxidative stress, and autophagosome formation [28]. Thus, we used cultured ECs and VSMCs to study the role of PHBs in AS in our *in vitro* experiments.

AS is now recognized as an inflammatory disorder of the arteries. Increasing evidence has suggested that cytokines are involved in AS, playing essential roles in its pathogenesis [29]. In

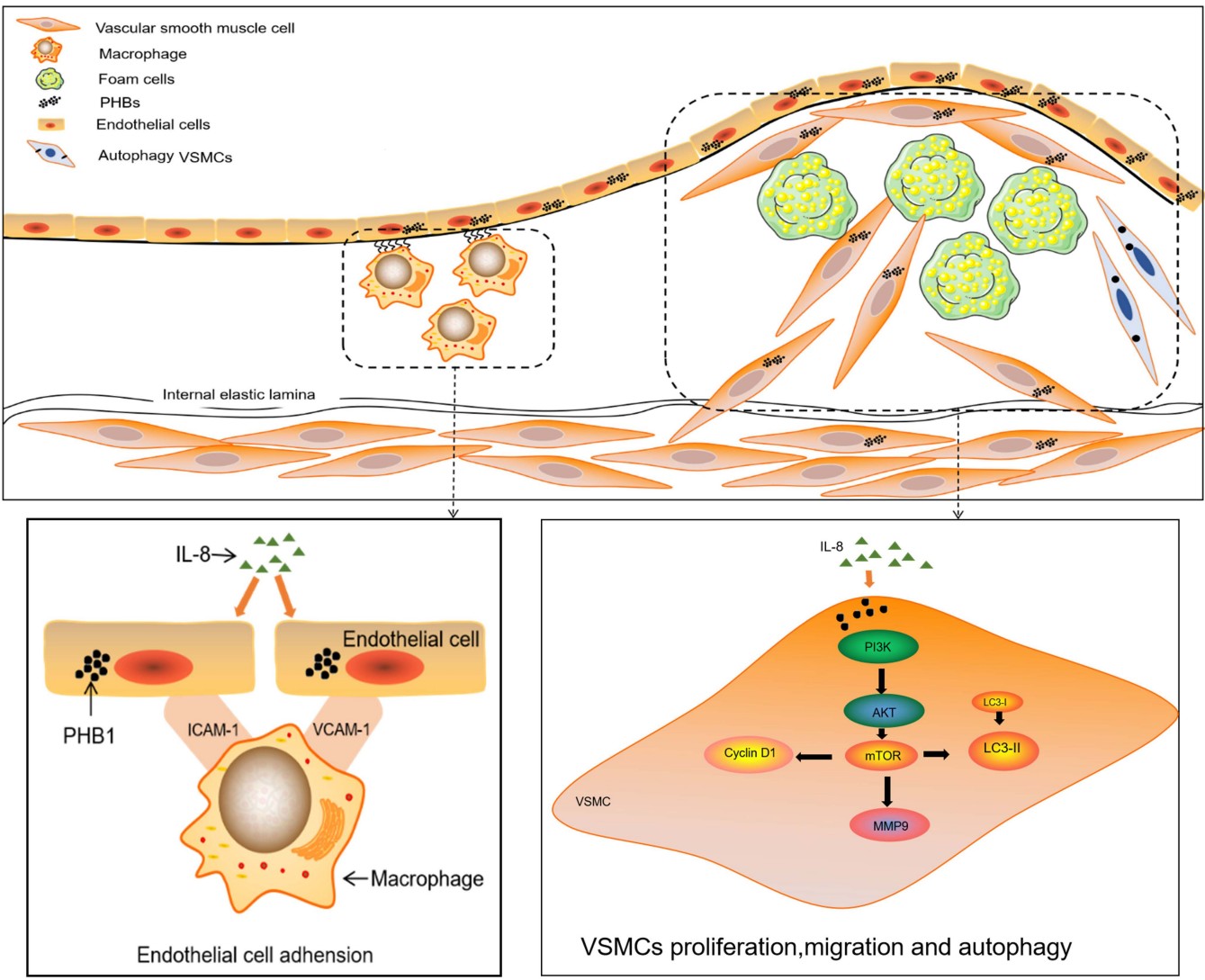

**Fig 9. A schematic diagram of PHB1 and PHB2 expression in blood vessel wall cells and the possible molecular pathways in terms of adhesion of ECs and proliferation, migration, and autophagy of VSMCs.**

particular, IL-8 induces the migration and contraction of airway smooth muscle cells [30], as well as the proliferation, migration, and apoptosis of ECs. In a previous study, IL-8 was highly expressed in lesion macrophages, ECs, and SMCs following the IL-8-induced establishment and preservation of the inflammatory-insulted vascular wall [31].

Increasing evidence has suggested the role of IL-8 in atherogenesis. Thus, we chose IL-8 to induce the adhesion of ECs as well as the proliferation, migration, and autophagy of VSMCs while exploring the function of PHB1/PHB2 and the involvement of signaling pathways in VSMCs. Importantly, IL-8 enhanced the adhesion of ECs and the migration and proliferation of VSMCs [32]. Subsequently, we detected the increased expression of PHB1/PHB2 in IL-8-induced ECs and VSMCs compared with those in the control group, in agreement with our earlier *in vivo* findings in mice. We further tested the effect of silencing PHBs and overexpressing PHB1 on vascular dysfunction in IL-8-induced cells. The abnormal proliferation and migration of VSMCs have been demonstrated in the pathological progression of AS [33]. Autophagy has been shown to play a vital role in the pathophysiological process in the vasculature and be involved in the development of AS [34]. We thus tested the role of endothelial dysfunction in AS progression. In our experiments, knocking down PHBs suppressed the adhesion of ECs and the proliferation and autophagy of VSMCs.These results demonstrated that the PHB1/PHB2 complex plays a crucial role in IL-8-induced vascular wall dysfunction (Fig 9).

The PI3K/Akt and MAPK pathways, critical prosurvival signaling pathways, reportedly play crucial roles in metabolism, apoptosis, and cell cycle [35]. Consequently, it has been proposed that the PI3K/Akt pathway might be relevant as a potential therapeutic target in treating AS.

In the present study, we investigated whether the PHB1/PHB2 complex affects the PI3K/Akt signaling. Our results showed that knocking down PHB1/PHB2 in IL-8-induced VSMCs inhibited the expression of p-Akt and p-mTOR, but not that of p-Erk1/2, which remained unchanged. SC79, an Akt agonist, was found to induce the activity of both p-Akt and p-mTOR. After Akt activation, the ability of PHB1/PHB2-downregulated cells to undergo proliferation, migration, and autophagy was significantly enhanced. Taken together, our findings indicated that PHB1/PHB2 influences the proliferation, migration, and autophagy of VSMCs by affecting the PI3K/AKT/mTOR pathway. However, this study still has some limitations. On one hand, THP-1 cells cannot strictly be considered monocytes, and future studies should continue to verify the function of PHB after inducing differentiation in THP-1 cells [36]. On the other hand, exploring changes in PHB protein or its localization through modulation of the AKT signaling pathway could be a valuable direction for future research [37,38]. Additionally, some studies have reported that activation of AKT with the AKT agonist SC79 should inhibit autophagy. Although this is based on rat oligodendrocyte precursor cells, it reflects the complexity of the AKT regulatory model [39], and we still need to refine the mechanistic pathways in future research.

In conclusion, we demonstrated that PHB1/PHB2 was upregulated in AS tissues and cell lines. Knocking down PHB1/PHB2 inhibited the adhesion of monocytes to ECs, as well as the proliferation, migration, and autophagy of IL-8-induced VSMCs. Mechanistically, the function of PHB1/PHB2 was mediated by the PI3K/Akt signaling pathway. Collectively, our study provides new insights regarding the use of PHB1/PHB2 as a potential therapeutic target for the treatment of AS.

## Supporting information

**S1 Fig. The expression of PHB1 was increased in the aortic vessel wall of *ApoE*$^{-/-}$ mice fed a high-fat diet (HFD).** (A) Representative immunohistochemistry images of α-SMA (a marker of VSMCs) and PHB1 in *ApoE*$^{-/-}$ mice fed an HFD and regular chow (RC), Brown

color represents positive cells; (B) Representative immunofluorescence images of α-SMA and PHB1 in *ApoE*⁻/⁻ mice with HFD and RC; scale bars: 200 μm.
(TIF)

**S2 Fig. The expression of PHB2 was increased in the aortic vessel wall of *ApoE*⁻/⁻ mice with HFD.** (A) Representative immunohistochemistry images of α-SMA (a marker of VSMCs) and PHB2 in *ApoE*⁻/⁻ mice with HFD and RC; (B) Representative immunofluorescence images of α-SMA and PHB2 in *ApoE*⁻/⁻ mice with HFD and RC; scale bars: 200 μm.
(TIF)

**S3 Fig. Expression of PHB1 and PHB2 in blood vessel wall cells.** (A) Western blotting showing the expression of PHB1 and PHB2 in HUVEC, A7r5, and AosMC cells treated with IL-8 (0–100 ng/mL). (B) Western blotting showing the expression of PHB1 and PHB2 in HUVEC, A7r5, and AosMC cells treated with IL-8 (50 ng/L) (0–48 h). All data are expressed as the mean ± SD (n = 5). *$p < 0.05$, **$p < 0.01$, ***$p < 0.001$.
(TIF)

**S4 Fig. Knockdown and overexpression of PHB1 in HUVEC.** (A) Western blotting confirmed the efficiency of siPHB1 in HUVECs; siPHB1 represents siRNA-PHB1. NC represents negative siRNA used as control. (B) Western blotting confirmed the efficiency of the pex-4-PHB1 plasmid in HUVECs. Pex-4 represents the empty vector, whereas PHB1 represents the pex-4-PHB1 plasmid. All data are expressed as the mean ± SD (n = 5). *$p < 0.05$, **$p < 0.01$, ***$p < 0.001$.
(TIF)

**S5 Fig. The effects of si-PHB1 on the activity of Erk and the p-c-Raf upstream protein.** Western blotting showing the expression of p-c-Raf (S295), p-ERK1/2 and ERK1/2. All data are expressed as the mean ± SD (n = 5). *$p < 0.05$, **$p < 0.01$, ***$p < 0.001$.
(TIF)

## Author contributions

**Conceptualization:** Xinxin Hu.

**Formal analysis:** Ying Cui, Yuanhua Qin.

**Project administration:** Mei Li, Xiaoyan Hu.

**Resources:** Xiaohua An.

**Validation:** Xiaoqing Wei, Ying Zhao.

**Visualization:** Fuhua Gao.

**Writing – original draft:** Mei Li, Xiaoyan Hu.

**Writing – review & editing:** Ying Gao.

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
