## [Decision Letter · Decision Letter 0]

14 Nov 2024

PONE-D-24-46813Inhibition of PHB1/PHB2 suppresses atherosclerotic plaque formation by interrupting PI3K/AKT/mTOR signalingPLOS ONE

Dear Dr. Gao,

Thank you for submitting your manuscript to PLOS ONE. After careful consideration, we feel that it has merit but does not fully meet PLOS ONE’s publication criteria as it currently stands. Therefore, we invite you to submit a revised version of the manuscript that addresses the points raised during the review process.

We look forward to receiving your revised manuscript.

Kind regards,

Alexis G. Murillo Carrasco

Academic Editor

PLOS ONE

Journal Requirements:

3. Funding Information and Financial Disclosure sections do not match:

We note that the grant information you provided in the ‘Funding Information’ and ‘Financial Disclosure’ sections do not match. 

 National Natural Science Foundation of China (No. 30470394, 31070719, 31370800, and 81402916).  

5. We note you have included a table to which you do not refer in the text of your manuscript. Please ensure that you refer to Table 1 in your text; if accepted, production will need this reference to link the reader to the Table.

Reviewers' comments:

Reviewer's Responses to Questions

**Comments to the Author**

1. Is the manuscript technically sound, and do the data support the conclusions?

Reviewer #1: Yes

Reviewer #2: Yes

Reviewer #3: Yes

Reviewer #4: Partly

2. Has the statistical analysis been performed appropriately and rigorously? 

Reviewer #1: Yes

Reviewer #2: Yes

Reviewer #3: Yes

Reviewer #4: Yes

3. Have the authors made all data underlying the findings in their manuscript fully available?

Reviewer #1: Yes

Reviewer #2: Yes

Reviewer #3: Yes

Reviewer #4: Yes

4. Is the manuscript presented in an intelligible fashion and written in standard English?

Reviewer #1: No

Reviewer #2: No

Reviewer #3: Yes

Reviewer #4: No

5. Review Comments to the Author

Reviewer #1: In the present manuscript, Li and colleagues showed that PHB1/PHB2 levels were elevated in serum of patients with hyperlipidemia. They further explored the effect of PHB1/PHB2 inhibition in the ApoE-/- fed with HFD murine model. They demonstrated that inhibition of PHB1 or PHB2 reduces the atherosclerosis progression. Mechanistically, they found that downregulating PHB1/PHB2 expression under inflammatory stimulation reduced the adhesion, proliferation, migration, and autophagy of ECs and VSMCs by inhibiting the PI3K/Akt/mTOR pathway activation. They have concluded that PHB1/PHB2 may play an important role in the promotion of endothelial and vascular smooth muscle cell injury. Thus, they could serve as a therapeutic target for atherosclerosis.

This finding is novel. Data are presented with qualitative and quantitative analysis; which is convincing. The whole manuscript could have been written in a more concise manner. Many methods/approaches/observations providing the same information repetitively appear in different sections throughout the manuscript. Extensive edition on the manuscript writing is warranted.

Here are two suggestions to scientifically improve the manuscript.

1. PHB1 and PHB2 are physically associated and functionally related proteins. It would be interesting to see the effect of double knockdown PHB1 and PHB2 in the AS progression. Are they functionally redundant?

2. Could you please provide a deeper discussion on the significance of the finding? Including The current knowledge on PHB1/PHB2 on VSMCs and ECs, future directions beyond this manuscript.

Minor points:

1. In figure 4B, the representative images and quantitative data do not indicate that overexpression of PHB1 increases THP-1 cell adhesion in the presence of il-8, as stated in Line 416

2. The order of Fig 4A and 4B is reversed, as the findings of VCAM-1 and ICAM-1 come later in the manuscript. (line 419)

Reviewer #2: This study reports that PHB1/PHB2 are up-regulated in AS tissues and cell lines. The knockdown of PHB1/PHB2 inhibits the adhesion of monocytes to ECs as well as proliferation, migration, and autophagy of IL-8-induced VSMCs, which might be mediated by the PI3K/Akt signaling pathway.The authors’ work represents PHB1/PHB2 as a potential therapeutic target for the treatment of AS in the clinical setting. The experiments are informative and sand the results are persuasive. However, there are still several comments that should be addressed:

1.In Figure 2D, hematoxylin and eosin staining and Masson’s staining of aortic plaques cannot reflect activity, please modify.

2.The confocal microscopy for PHB in Figure 3D appears to be located in organelles. They are not diffused in the cytosol and more likely punctalized.

3.The protein expression level in Figure 4A is not significant, please reassign the picture.

4.THP-1 is not strictly a monocyte.It should be induced to differentiate into monocytes to enhance the persuasiveness of experiments.

5.Why choose MMP9 over other members of MMPs? Moreover, Gelatin Zymography method should also be used to detect the activity of MMP9 for the reason that Western blotting cannot demonstrate that PHB1 and PHB2 inhibit IL-8-induced VSMC migration.

6.When the Akt downstream signaling pathway was activated, phosphorylated Akt levels were increased and both PHB1 and PHB2 were up-regulated compared to IL-8-treated A7r5 cells. If the AKT signaling pathway is down-regulated, will the levels of both PHB1 and PHB2 also decrease?

Minor:

1.It will be more convincing if the clinical cases can be more.

2.In Figure S3 B,why is there no picture that show the protein level of PHB2 in AoSMC ?

3.In Figure 4, there is no statistical significance in OE-PHB1.In addition, why is there no PHB2?

Reviewer #3: Dear Editor/Authors,

I am writing to inform you that I have completed my review of the manuscript titled " Inhibition of PHB1/PHB2 suppresses atherosclerotic plaque formation by interrupting PI3K/AKT/mTOR signaling” and I am pleased to accept it for publication in its current form. After careful consideration, I have found the manuscript to be well-written, well-structured, and of high quality. The authors have presented their research in a clear and concise manner, and the manuscript meets all the journal's requirements.

I have recommended minor revisions mentioned in PDF file as comment, and I believe the manuscript should be accepted for publication. I have checked the manuscript for its originality, relevance, and impact, and I am confident that it will make a significant contribution to the field.

Please let me know if you need any further information from me. I appreciate the opportunity to review this manuscript and look forward to seeing it in print.

Best regards,

Muhammad Asmat Ullah Saleem

Reviewer #4: Major Comments:

Please cite the relevant figures for lines 353-355, 362-365, and 380-382 (show data supporting PHB localization in the cytoplasm).

Please comment on why PHB1/PHB2 levels are lower at 100 ng/mL in Fig S3—is this caused by cell death?

Does Figure 3D correspond to stimulated or unstimulated conditions? How does the localization of PHB1 and PHB2 vary under both conditions?

Please discuss the text in the order that the results are displayed—figure 3D is discussed before 3C.

The legend for figures 3C and 3D are swapped.

How is pex-PHB1 overexpressed? Please elaborate in the methods section.

Figure 4 and its legend are confusing. Significant improvement in clarity is needed for figure 4.

1) What are the two controls in 4A? The legends for 4A and 4B are flipped.

2) No obvious change in ICAM1 and VCAM1. Confirmation using alternative methods such as immunofluorescence will further boost confidence.

3) Does overexpression of PHB1 increase ICAM1 and VCAM1 expression levels using western blot in figure 4? Please

show.

For autophagy experiments, it is important to confirm the flux of autophagy to make sure the increase in autophagosomes is a result of autophagy activation and not due to blockage of clearance. Please see doi: 10.1080/15548627.2020.1797280 on how to measure flux.

Need more text elaborating on the findings in figures 8H, I, J, K, and L.

Activation of the AKT-mTOR axis as shown in figures 8A, B, C, and D should inhibit autophagy, but the results claim otherwise. Please address this discrepancy.

Activation of AKT using the AKT agonist SC79 should inhibit autophagy (see DOI: 10.1007/s11064-023-04057-w). The results appear contradictory; please address this.

Minor Comments:

In lines 102-103, should it be “IL-8 inducing AS” and not the other way around?

Color code aSMA, PHB1, and DAPI for better clarity in figure S1B.

Please point out the specific changes with arrows to guide the audience in S1.

Please include a scale bar on the last panel of figure S1B.

Please cite literature for lines 374-375, 458-459, and 469.

Lines 359-361 can be structured better for clarity. Both uninfected and infected mice were fed with a normal diet, correct? The main difference is whether they are infected or not.

Please mention each abbreviation used in the figure in the legend for all figures.

Figures 7C and 7D: the y-axis label could be more accurate. Does it indicate expression of the protein relative to alpha-tubulin?

For figure 8B, are GAPDH levels measured? Please check the y-axis.

Please mention where SC79 is sourced from in the materials/methods section. Please check grammar and improve clarity.

6. PLOS authors have the option to publish the peer review history of their article (what does this mean? ). If published, this will include your full peer review and any attached files.

**Do you want your identity to be public for this peer review?** For information about this choice, including consent withdrawal, please see our Privacy Policy .

Reviewer #1: **Yes: ** Ying Wang

Reviewer #2: **Yes: ** Wuyang Wang

Reviewer #3: **Yes: ** Muhammad Asmt Ullah Saleem

Reviewer #4: No

---

## [Author Response · Author response to Decision Letter 1]

24 Dec 2024

Dear editors and reviewers,

We are truly grateful to the comments and suggestions from your review. Based on these comments and suggestions, we have made careful modifications on the manuscript. All the changes made to the text are in red color. You will find our point by point responses to the comments as bellow.

Journal Requirements:

Response: Thank you for providing the PLOS ONE style templates. We have carefully reviewed and revised our manuscript to fully comply with the style requirements, including file naming conventions as outlined. We have utilized both provided templates to ensure our submission meets all specified formatting guidelines.

Response: We appreciate the reminder regarding the ORCID iD requirement. I have ensured that my ORCID iD is linked and validated in Editorial Manager.

3. Funding Information and Financial Disclosure sections do not match:

We note that the grant information you provided in the ‘Funding Information’ and ‘Financial Disclosure’ sections do not match.

Response: Thank you for pointing out the discrepancies in the 'Funding Information' and 'Financial Disclosure' sections. We have carefully revised these sections to ensure that all grant numbers are correctly listed and consistent across both sections, reflecting the accurate funding sources for our study.

National Natural Science Foundation of China (No. 30470394, 31070719, 31370800, and 81402916).

Response: We have reviewed the role of our funders and confirm that the National Natural Science Foundation of China (Nos. 30470394, 31070719, 31370800, and 81402916) had no role in the study design, data collection and analysis, decision to publish, or preparation of the manuscript. We have revised "Funding section" information in manuscript.

5. We note you have included a table to which you do not refer in the text of your manuscript. Please ensure that you refer to Table 1 in your text; if accepted, production will need this reference to link the reader to the Table.

Response: Thank you for noting the omission of a reference to Table 1 in the text of our manuscript. We have now included specific references to Table 1 within the relevant sections of the manuscript to ensure that readers can clearly connect the text discussions to the data presented in the table.

Responses to the comments from Reviewer 1

1. PHB1 and PHB2 are physically associated and functionally related proteins. It would be interesting to see the effect of double knockdown PHB1 and PHB2 in the AS progression. Are they functionally redundant?

Response: We greatly appreciate your suggestion and agree that the functionality of the PHB1 and PHB2 proteins is intriguing. Indeed, PHB1 and PHB2 can form a complex within the mitochondrial inner membrane, serving a chaperone-like function for newly synthesized mitochondrial proteins. These homologs are interdependent: the absence of one subunit leads to instability in the other, indicating that the PHB complex represents an active structural unit. Previous studies indicate that, particularly in lower organisms, PHB1 predominantly exerts influence (Schleicher et al., 2008; Tatsuta et al., 2005). Furthermore, in Figure 5F, we conducted a double knockdown experiment, observing a more pronounced reduction in cell proliferation compared to single knockdowns. Additionally, in Figure 3C, we demonstrated an interaction between the two proteins. Thus, we argue that they function collaboratively, a point we have expanded upon in the discussion section of our manuscript.

Figure 5F. Cell proliferation was measured using the EdU assay; scale bars: 200 μm.

2. Could you please provide a deeper discussion on the significance of the finding? Including The current knowledge on PHB1/PHB2 on VSMCs and ECs, future directions beyond this manuscript.

Response: Thank you for encouraging a more detailed discussion on the significance of our findings. We have expanded our discussion to include a broader context of PHB1 and PHB2's roles in vascular smooth muscle cells (VSMCs) and endothelial cells (ECs), referencing existing literature to underscore their emerging significance in vascular pathologies. We also propose future research directions, highlighting the potential translational impact of targeting these proteins in therapeutic strategies for atherosclerosis.

Minor points:

1. In figure 4B, the representative images and quantitative data do not indicate that overexpression of PHB1 increases THP-1 cell adhesion in the presence of il-8, as stated in Line 416

Response: We appreciate your feedback pointing out the error in our description, and we have made the necessary corrections. Upon further analysis, we indeed found that the overexpression of PHB1 does not significantly affect cell adhesion.

2. The order of Fig 4A and 4B is reversed, as the findings of VCAM-1 and ICAM-1 come later in the manuscript. (line 419)

Response: We apologize for the oversight and have corrected the text accordingly. We appreciate your swift action to rectify the sequence of descriptions and figure legends to better match the corresponding figures.

Responses to the comments from Reviewer 2

1.In Figure 2D, hematoxylin and eosin staining and Masson’s staining of aortic plaques cannot reflect activity, please modify.

Response: We appreciate your attention to detail, and we have reviewed the concerns you raised. Indeed, we noticed that Figure 2C involves H&E and Masson's trichrome staining, and we used the quantification of positive areas rather than activity. Figure 2D, featuring immunohistochemical staining for Cyclin D1, MMP-9, and LC3, uses "IHC" to denote the presence of positive signals—a common terminology in this context. We hope this clarifies the matter and look forward to your understanding.

Figure 2. Effects of lentiviral carriers in ApoE-/- mice.

2.The confocal microscopy for PHB in Figure 3D appears to be located in organelles. They are not diffused in the cytosol and more likely punctalized.

Response: We appreciate your point and thank you for your correction. Indeed, PHB proteins function as part of a complex in mitochondrial formation, and previous studies have also reported occasional nuclear localization, aligning with our immunofluorescence results(Schleicher et al., 2008). Therefore, we have corrected the inaccurate description in the text, confirming that PHB proteins, similar to previous reports, are predominantly located in the mitochondrial compartments within the cytoplasm.

3.The protein expression level in Figure 4A is not significant, please reassign the picture.

Response: Thank you for your feedback. We have corrected the sequence of images in Figures 4A and 4B to accurately reflect the findings discussed in the text. The quantitative analysis based on original grayscale measurements remains unchanged, ensuring that the conclusions drawn are supported by authentic data. We appreciate your suggestion, which has helped us improve the clarity of our presentation.

4.THP-1 is not strictly a monocyte.It should be induced to differentiate into monocytes to enhance the persuasiveness of experiments.

Response: We greatly appreciate your feedback. Indeed, we have noted that there has been discussion in the literature about the confusing use of THP-1 cells as a model for monocytes(Bosshart and Heinzelmann, 2016). Due to current experimental system limitations, we are unable to conduct induced experimental systems at this time. We will address this limitation in the discussion section and aim to optimize future studies. We hope for your understanding in this matter.

5.Why choose MMP9 over other members of MMPs? Moreover, Gelatin Zymography method should also be used to detect the activity of MMP9 for the reason that Western blotting cannot demonstrate that PHB1 and PHB2 inhibit IL-8-induced VSMC migration.

Response: We appreciate your interest in the verification aspects of our study. There are two main reasons for selecting MMP9: firstly, previous literature has reported an association between PHB and certain proteins in the MMP family, such as MMP2 and MMP9(Dong et al., 2016; Wang et al., 2015). Additionally, our laboratory has found that the MMP9 antibody is suitable for specific staining and Western blotting to reflect phenotypic changes. Furthermore, MMP9 is a crucial downstream effector in the AKT pathway, reinforcing its relevance to our study. Other antibodies in the same family, such as MMP3 and MMP10, have shown non-specific staining in our tests. On the other hand, due to limitations in our experimental platform and budget, we are currently unable to use the Gelatin Zymography method for further verification. We hope for your understanding on this matter, and we will discuss these experimental design limitations in the discussion section of our manuscript.

6.When the Akt downstream signaling pathway was activated, phosphorylated Akt levels were increased and both PHB1 and PHB2 were up-regulated compared to IL-8-treated A7r5 cells. If the AKT signaling pathway is down-regulated, will the levels of both PHB1 and PHB2 also decrease?

Response: Thank you for your insightful question regarding the relationship between Akt signaling and PHB1/PHB2 levels. We share your interest in this area, but currently, we lack the inhibitors necessary to test the effects of down-regulating the Akt pathway on PHB1 and PHB2. Consequently, we cannot provide direct experimental evidence at this moment. However, we plan to explore this further in future studies as we acquire the necessary resources to conduct these experiments. We appreciate your understanding and are committed to continuing this line of research to deepen our understanding of these interactions

Minor:

1.It will be more convincing if the clinical cases can be more.

Response: Thank you for your kind suggestion. Due to hospital platform constraints, the number and enrollment of patients are limited, and we cannot significantly increase the sample size in a short period. We plan to continue collecting blood and even tissue samples from patients in the future to further validate our findings. We appreciate your understanding in this regard.

2.In Figure S3 B,why is there no picture that show the protein level of PHB2 in AoSMC ?

Response: Thank you for your insightful question, the western blotting detection of PHB2 in AoSMC showed faint bands with prolonged exposure. We believe that the quantification of such vague bands could significantly affect the accuracy of the results; therefore, we did not include this data in our study.

3.In Figure 4, there is no statistical significance in OE-PHB1.In addition, why is there no PHB2?

Response: We appreciate your feedback pointing out the error in our description, and we have made the necessary corrections. Upon further analysis, we indeed found that the overexpression of PHB1 does not significantly affect cell adhesion. We have focused on the role of PHB1 in our experiments because, as observed in Figure 3C, PHB1 and PHB2 may function as a complex. Therefore, we prioritized the knockdown of PHB1 to understand its impact, which is sufficient for the scope of this study. Thank you for your understanding.

Responses to the comments from Reviewer 3

Thank you for your kind acknowledgment of our manuscript. We are committed to integrating all reviewers' suggestions to enhance our submission further, ensuring it meets the scientific standards of the journal and its readership. We appreciate the opportunity to refine our work based on the feedback provided.

Responses to the comments from Reviewer 4

Please cite the relevant figures for lines 353-355, 362-365, and 380-382 (show data supporting PHB localization in the cytoplasm).

Response: We appreciate your guidance on specifying PHB localization in the cytoplasm. In lines 411-413 of our manuscript, we have cited the immunohistochemistry figures to illustrate this point, and subsequently referenced Figure 3D. We hope this clarification meets your understanding.

Please comment on why PHB1/PHB2 levels are lower at 100 ng/mL in Fig S3—is this caused by cell death?

Response: We appreciate your interest in the specific dosage used in our experiments. Indeed, we found 50 ng/ml of IL-8 to be a suitable concentration for modeling, as you indicated. However, at 100 ng/ml, there was a decrease in PHB expression, but without testing apoptosis-related markers, we cannot definitively conclude that apoptosis caused these changes. Given that IL-8 is a potent angiogenic factor that can influence the migration and proliferation of endothelial and smooth muscle cells and may play a role in atherosclerosis, overly high concentrations might inherently alter cell functions. To avoid confusing or misleading our conclusions, we hope you understand why we cannot definitively attribute the observed expression changes to apoptosis caused by excessive IL-8 levels in this manuscript.

Does Figure 3D correspond to stimulated or unstimulated conditions? How does the localization of PHB1 and PHB2 vary under both conditions?

Response: We appreciate your interest in the specific details of the immunofluorescence method described in Figure 3D. As mentioned in the methods section, this staining was performed on unstimulated cells to demonstrate that PHB proteins primarily localize to the mitochondrial compartments within the cytoplasm. PHB proteins function as part of a complex essential for mitochondrial formation, and as such, they do not exhibit significant nuclear translocation as functional proteins. The immunofluorescence presents a punctate pattern, indicative of PHB's role in mitochondrial function, rather than focusing on protein localization changes, which would be challenging to align with subsequent analysis logic. We hope you understand our reasoning and the focus of our experimental design.

Please discuss the text in the order that the results are displayed—figure 3D is discussed before 3C.

Response: We appreciate your diligence in pointing out the discrepancies in our manuscript's sequence descriptions. We have made the necessary corrections to the manuscript to ensure accuracy and consistency. Thank you for your attention to detail, which helps improve the quality of our work.

The legend for figures 3C and 3D are swapped.

Response: We appreciate your diligence in pointing out the discrepancies in Figure legend. We have made the necessary corrections to the manuscript to ensure accuracy and consistency.

How is pex-PHB1 overexpressed? Please elaborate in the methods section.

Response: Thank you for highlighting the oversight in our methodological description. We have now updated the Methods section to include the construction of the PEX-PHB1 vec

---

## [Decision Letter · Decision Letter 1]

20 Feb 2025

Inhibition of PHB1/PHB2 suppresses atherosclerotic plaque formation by interrupting PI3K/AKT/mTOR signaling

PONE-D-24-46813R1

Dear Dr. Gao,

We’re pleased to inform you that your manuscript has been judged scientifically suitable for publication and will be formally accepted for publication once it meets all outstanding technical requirements.

Kind regards,

Alexis G. Murillo Carrasco

Academic Editor

PLOS ONE

Additional Editor Comments (optional):

Reviewers' comments:

Reviewer's Responses to Questions

**Comments to the Author**

1. If the authors have adequately addressed your comments raised in a previous round of review and you feel that this manuscript is now acceptable for publication, you may indicate that here to bypass the “Comments to the Author” section, enter your conflict of interest statement in the “Confidential to Editor” section, and submit your "Accept" recommendation.

Reviewer #1: All comments have been addressed

2. Is the manuscript technically sound, and do the data support the conclusions?

Reviewer #1: Yes

3. Has the statistical analysis been performed appropriately and rigorously? 

Reviewer #1: I Don't Know

4. Have the authors made all data underlying the findings in their manuscript fully available?

Reviewer #1: Yes

5. Is the manuscript presented in an intelligible fashion and written in standard English?

Reviewer #1: No

6. Review Comments to the Author

Reviewer #1: The quality of revised manuscript has been significantly improved. Minor English editing is needed to make it more concise and informative.

---

## [Editor Report · Acceptance letter]

PONE-D-24-46813R1

PLOS ONE

Dear Dr. Gao,

I'm pleased to inform you that your manuscript has been deemed suitable for publication in PLOS ONE. Congratulations! Your manuscript is now being handed over to our production team.

Kind regards,

on behalf of

Dr. Alexis G. Murillo Carrasco

Academic Editor

PLOS ONE